# Extraction, Characterization, and Platelet Inhibitory Effects of Two Polysaccharides from the Cs-4 Fungus

**DOI:** 10.3390/ijms232012608

**Published:** 2022-10-20

**Authors:** Yu-Heng Mao, Feng-Lin Song, Yi-Xuan Xu, Ang-Xin Song, Zhao-Mei Wang, Ming-Zhu Zhao, Fang He, Ze-Zhong Tian, Yan Yang

**Affiliations:** 1School of Public Health (Shenzhen), Shenzhen Campus of Sun Yat-sen University, Sun Yat-sen University, Shenzhen 518107, China; 2School of Exercise and Health, Guangzhou Sport University, Guangzhou 510500, China; 3School of Food Science, Guangdong Pharmaceutical University, Guangzhou 510006, China; 4Guangdong Provincial Key Laboratory of Food, Nutrition and Health, School of Public Health, Sun Yat-sen University (Northern Campus), Guangzhou 510080, China; 5Guangdong Engineering Technology Research Center of Nutrition Translation, Guangzhou 510080, China; 6School of Liquor and Food Engineering, Guizhou University, Guiyang 550025, China; 7School of Food Science & Engineering, South China University of Technology, Guangzhou 510641, China

**Keywords:** *Ophiocordyceps sinensis*, *Cordyceps sinensis*, cardiovascular disease, antiplatelet, thrombosis, microbial metabolites, SCFAs

## Abstract

Cardiovascular diseases are associated with platelet hyperactivity, and downregulating platelet activation is one of the promising antithrombotic strategies. This study newly extracted two polysaccharides (purified exopolysaccharides, EPSp and purified intercellular exopolysaccharides, IPSp) from *Cordyceps sinensis* Cs-4 mycelial fermentation powder, and investigated the effects of the two polysaccharides and their gut bacterial metabolites on platelet functions and thrombus formation. EPSp and IPSp are majorly composed of galactose, mannose, glucose, and arabinose. Both EPSp and IPSp mainly contain 4-Gal*p* and 4-Glc*p* glycosidic linkages. EPSp and IPSp significantly inhibited human platelet activation and aggregation with a dose-dependent manner, and attenuated thrombus formation in mice without increasing bleeding risk. Furthermore, the EPSp and IPSp after fecal fermentation showed enhanced platelet inhibitory effects. The results have demonstrated the potential value of Cs-4 polysaccharides as novel protective ingredients for cardiovascular diseases.

## 1. Introduction

Arterial thrombotic diseases are the leading cause of morbidity and mortality worldwide [1]. Platelets play important roles in hemostasis and thrombosis [1]. On the one hand, platelet activation and aggregation at the sites of vascular injury are key events required for hemostasis [2]. On the other hand, the excessive accumulation or abnormal hyperactivity may lead to thrombotic diseases [2]. Many cardiovascular diseases (CVDs) are associated with platelet hyperactivity, while the overactivation of platelets is also recognized as an independent risk factor for CVDs [1].

Platelet activation induced by platelet agonists at sites of the vessel wall injury is an important process for both protective hemostasis and pathological thrombosis. Following activation, platelet granule contents and cytosol proteins (such as P-selectin) are secreted into the circulation or translocate to the cell surface, and the αIIbβ3 integrin may also converse to an active conformation that enables αIIbβ3 integrin to bind ligands including fibrinogen and other proteins [3]. The translocation and conformation of these proteins following platelet activation happen throughout the entire cascade reaction and are important processes in hemostasis and thrombosis, which facilitate platelets to interact with their environment. Numerous studies have shown that downregulating platelet activation might be a promising approach for the prevention and treatment of CVDs [4,5]. However, the antiplatelet drugs (such as aspirin) clinically used may increase the bleeding risk [6], therefore, the natural antiplatelet ingredients with higher safety attracted increasing attention for improving cardiovascular health.

Natural polysaccharides from edible plants, fungi, and other resources have been widely developed as functional foods and therapeutic ingredients for their significant health benefits. Some polysaccharides were reported to show therapeutic potential for cardiovascular diseases [7,8], metabolic diseases [9,10], etc. However, only a few studies reported the effects of natural polysaccharides on platelet functions. A study reported that a polysaccharide extracted from *Cordyceps sinensis* mycelial fermentation (unannotated fungus) directly inhibited human platelet activation in vitro [11]. However, as most of these complex polysaccharides are non-digestible while commonly utilized by gut microbiota, their benefits were considerably attributed to their gut microbial metabolites, such as short chain fatty acids (SCFAs) [12]. Furthermore, a previous study showed that bifidobacterial fermentation dramatically enhanced the anti-inflammatory effects of an exopolysaccharide from a medicinal fungus Cs-HK1 liquid fermentation medium, and it might be due to the functional metabolites produced by gut bacteria [13].

The Chinese caterpillar fungus *Ophiocordyceps sinensis* (Berk.) (=*Cordyceps sinensis*, or Dong-Chong-Xia-Cao in Chinese) is a valuable edible and medicinal fungus with a wide range of therapeutic and health-promoting effects, including immunomodulatory, antioxidative, hepatoprotective, and neuroprotective activities [14]. As natural resource is sharply decreasing, *O. sinensis* mycelial fermentation has been widely applied in industry to produce *O. sinensis* fungal materials to meet the rising demand in the market. Cs-4 (*Paecilomyces hepiali*) is an officially verified fungus for commercial *O. sinensis* mycelial fermentation in China [14], and Cs-4 mycelial culture extracts were reported to show many health-improving effects, including anti-inflammation [15], skincare benefits [14], etc. Therefore, we hypothesized that Cs-4 polysaccharides could inhibit the hyperactivity of platelets, while the platelet-inhibiting effects could be further enhanced with gut bacterial fermentation. To the best of our knowledge, no studies were reported on the effects of Cs-4 polysaccharides and their gut bacterial metabolites on platelets.

This study was to evaluate the potential effects of two newly extracted Cs-4 polysaccharides and their gut microbial metabolites on platelets. The exopolysaccharides and intracellular polysaccharides were extracted by ethanol precipitation from the *Cordyceps sinensis* Cs-4 mycelium culture extracts and purified by deproteinization. The two Cs-4 polysaccharides and their metabolites (fecal fermentation medium liquid and short chain fatty acids) were assessed for their inhibitory effects on human platelet activation and aggregation in vitro, and further demonstrated for their effects on thrombus formation in vivo.

## 2. Results

### 2.1. Yields and Molecular Properties

As shown in Table 1, the yields of exopolysaccharides (EPS), intracellular polysaccharides (IPS), purified exopolysaccharides (EPSp), and purified intracellular polysaccharides (IPSp) were 6.95%, 3.65%, 2.87%, and 0.75% respectively. There were two major peaks for EPS, IPS, EPSp, and IPSp on the Gel Permeation Chromatography (GPC) as shown in Appendix A. The average molecular weight (MW) of EPS, IPS, and EPSp was in the same order of 10^4^ (major) and 10^3^ (minor), while the proportion of the lower MW peak in the order of 10^3^ was more than 50% in IPS. After purification, the protein content of EPS (3.93%) and IPS (4.57%) was decreased to 1.11% and 0.78%, respectively, while the carbohydrate content increased from 88.08% to 90.09% for EPS and 84.00% to 88.90% for IPS.

### 2.2. Methylation Analysis of EPSp and IPSp

As shown in Appendix A, the methylated monosaccharides and substitution positions are obtained. The methylation results showed that EPSp had six main linkages (relative molar ratio > 5%) including the 4-Gal*p* and 4-Glc*p* with relative molar ratio of 31.37% and 19.93%, while IPSp had four main linkages including the 4-Gal*p* and 4-Glc*p* with relative molar ratios of 19.00% and 27.94%.

### 2.3. Monosaccharide Composition

As shown in Table 2, the monosaccharide ratio of the EPSp and IPSp were similar in general. Both of EPSp and IPSp were composed of fucose, arabinose, galactose, glucose, xylose, mannose, and ribose, of which arabinose, galactose, glucose, and mannose were the major monosaccharides. These three monosaccharides were also common monosaccharides that compose other *Cordyceps sinensis* polysaccharides [13,16]. The chromatography profiles of EPSp and IPSp are shown in Appendix A.

### 2.4. Fourier Transform Infrared (FT-IR)

As shown in Figure 1, FT-IR spectra of EPSp and IPSp were nearly identical, which indicates the close similarity on the primary structure and functional groups between the two purified polysaccharides. Specifically, the large absorption peak at 3400 cm^−1^ and the weak absorption peak at 2910 cm^−1^ were attributed to the −OH stretching vibration and C−H stretching vibration, respectively. The absorption peak at 1600 cm^−1^ indicates the COO− stretching vibration, which might be attributed to the esterification of a few side chains or the existence of uronic acid. Moreover, several absorption peaks around 1000 cm^−1^ were attributed to the C−O−C bond vibration and the pyran configuration. Overall, the IR spectra indicated that EPSp and IPSp contain the major functional groups of sugars.

### 2.5. Scanning Electron Microscopy Observation

Figure 2 shows the microstructures of EPSp and IPSp. The EPSp with relatively higher MW appeared as an extreme thin sheet with holes and folds on it, while the IPSp with relatively lower MW appeared as cotton with short fibrous filaments on the edges. The different appearances of the two polysaccharides might be due to their different MW and molecular structures [17].

### 2.6. SCFA Production after Fecal Fermentation of EPSp and IPSp

Table 3 presents the concentrations of acetic acid, propionic acid, butyric acid, and valeric acid and the total SCFAs in the fecal fermentation samples. Fecal fermentation without polysaccharides increased acetic acid, butyric acid, valeric acid, and the total SCFAs significantly compared with the medium before fermentation. The fecal fermentation of EPSm and IPSm dramatically increased the production of four individual and total SCFAs (one-way analysis of variance, ANOVA, *p* < 0.05) compared with the basal medium before and after fecal fermentation, and no significant differences on the SCFA production are found between EPSpm and IPSpm.

### 2.7. The Effects of EPSp, IPSp and Their Fecal Fermentation Metabolites on Platelet Activation

CD62P (P-selectin) plays an important role in the adhesion and aggregation of platelets, and it is one of the most widely used markers indicating the degree of platelet activation. As shown in Figure 3a,b, EPSp and IPSp significantly inhibited the CD62P expression on platelet surfaces compared with Ctl. Furthermore, the gut microbial metabolites of EPSp and IPSp significantly inhibited the CD62P expression in a dose-dependent manner.

Integrin αIIbβ3 is the most abundant receptor on the surface of platelets and is required for platelet aggregation. Upon activation, αIIbβ3 changes the conformation, consequently inducing the ligand binding site from a low to a high-affinity state. PAC-1 is a monoclonal antibody that specifically binds to the activated form of integrin αIIbβ3, thus it is usually used to assess the platelet activation. Figure 3c,d shows that EPSp and IPSp significantly inhibited PAC-1 binding to platelets compared with Ctl, and their gut microbial metabolites further enhanced the effect on inhibiting platelet activation compared with Ctlm.

### 2.8. Inhibitory Effects of EPSp and IPSp on Human Platelet Aggregation

Figure 4 shows the effects of the two purified polysaccharides and their fecal bacterial metabolites on platelet aggregation. EPSp at the high concentration (final concentration of 1 mg/mL) showed significant anti-aggregation effect on human platelets (ANOVA, *p* < 0.05), while low concentration showed no significant effect (Figure 4a,c) compared with the control. IPSp at both low and high concentrations significantly inhibited platelet aggregation (Figure 4d,f) compared with the control.

Notably, after fecal bacterial fermentation, the supernatant containing bacterial metabolites of EPSp and IPSp dramatically (ANOVA, *p* < 0.05) inhibited the platelet aggregation induced by collagen (2 μg/mL) with a dose-dependent trend (Figure 4b,e) compared with the basal medium (without polysaccharides) either before or after fecal fermentation. Additionally, the unfermented medium showed no effect on platelet aggregation, while the basal medium after fecal fermentation significantly decreased the aggregation rate compared with the control and unfermented medium (Figure 4b,c,e,f).

As SCFAs are the major metabolites of Cs-4 polysaccharides by gut bacteria, the effects of four major SCFAs on platelet aggregation were tested further. As shown in Appendix A, the four individual SCFAs showed significant effects on inhibiting the platelet aggregation induced by collagen in a dose-dependent manner (ANOVA, *p* < 0.05). Additionally, the platelet activation was also dose-dependently inhibited by the four individual SCFAs (Appendix A, ANOVA, *p* < 0.05). The intervention concentrations of the individual SCFAs were decided to be 20, 40, and 60 μg/mL after comprehensive consideration on the wide range of human serum concentration (0.1–60 μg/mL) [18]. Furthermore, the anti-aggregation effects of EPSp, IPSp, fecal microbial fermentation supernatant, and individual SCFAs on platelet rich plasma (PRP) were further demonstrated. Additionally, all polysaccharides, SCFAs and fecal microbial fermentation supernatant showed no significant influence on the viability of platelets (Appendix A). The results showed that the trends of the effects in more complex environment were similar to the gel-filtered platelets (Appendix A).

### 2.9. The Effects of EPSp, IPSp and Their Fecal Fermentation Metabolites on Bleeding Time in Mice

Figure 5 and Appendix A showed that both the two purified polysaccharides and their gut bacterial fermentation supernatants in the experimental concentrations had no significant influence on the bleeding time of mice. This indicated that all the interventions were safe and did not increase bleeding risk.

### 2.10. Anti-Thrombosis Effects of EPSp, IPSp and Their Fecal Fermentation Metabolites in Mice

As shown in Figure 6a, in untreated mice, multiple thrombi more than 20 μm are observed in the mesenteric arteriole at 10 min after FeCl_3_ injury. The time to the 1st thrombus in all untreated mice was within 5 min after FeCl_3_ injury (n = 6). In contrast, one bolus injection of platelets pre-incubated with EPSp, IPSp, EPSpm, and IPSpm before FeCl_3_ injury significantly prevented thrombus formation over 5 min (Figure 6a), and EPSpm and IPSpm extremely significantly (*p* < 0.001, *t*-test) prolonged the time to 1st thrombus formation (>20 μm) compared with the mice receiving the Ctlm (Figure 6b). The number of emboli (>20 μm) in 20 min and the size of the emboli in the 20 min from the time of the first thrombus were all significantly (*p* < 0.01, *t*-test) inhibited by EPSpm and IPSpm (Figure 6c,d). All together, these data confirm the in vivo inhibitory effects of EPSp, IPSp, EPSpm, and IPSpm on thrombus formation. Additionally, acetic acid, propionic acid, and butyric acid also significantly attenuated the thrombus formation in mice (Appendix A).

## 3. Discussion

*Cordyceps* mycelial fermentation products are ideal replacements for meeting the growing requirement of natural *Cordyceps sinensis*. As one of the main functional components in *Cordyceps sinensis*, the polysaccharides extracted from *Cordyceps* mycelial fermentation also attracted increasing attention. The MW of *C. sinensis* polysaccharides varied from several thousands to millions Daltons among different fungus strains [19]. The MW of an exopolysaccharide extracted from a medicinal fungus (Cs-HK1) liquid fermentation reached 1.93 × 10^8^ Da in our previous study [20], while the MW of a polysaccharides (CME-1) isolated from an unannotated *C. sinensis* mycelia was only 2.76 × 10^4^ Da [11]. In the present study, the MW of the two Cs-4 polysaccharides was 2.83 × 10^4^ Da and 7.36 × 10^3^ Da (weighted averages of all peaks in GPC profiles), respectively, which was significantly lower than Cs-HK1 EPS and similar with CME-1. The MW was closely related with the fermentability of polysaccharides, and the higher MW of polysaccharides usually resulted in a slower fermentation rate by gut bacteria [21]. Therefore, the significant increase of SCFA production might be due to the moderate MW of Cs-4 EPSp and IPSp. Except the moderate MW, the porous and fibrous microstructure of the two polysaccharides observed by scanning electron microscopy (SEM) also facilitated their utilization by gut bacteria [21].

Besides the MW and morphology, the molecular chemical structure was also essential to the biological functions of polysaccharides. Arabinose, mannose, glucose, and galactose are the major monosaccharides detected in the two Cs-4 polysaccharides, and very small amounts of fucose, xylose, and ribose were found in IPSp (Appendix A). These four major monosaccharides were usually found in natural or cultural *Cordyceps sinensis* polysaccharides, but with different molar ratios. Methylation analysis showed that 4-Gal*p* and 4-Glc*p* accounted for approximately 50% of the glycosidic linkages for the two Cs-4 polysaccharides. Additionally, the two polysaccharides had a considerable amount of t-Man*p* (>10%). The previous study reported that CME-1 was mainly composed of mannose and galactose at a ratio of 4:6, and the backbone of CME-1 was identified as repeated (1→4)-β-D-mannopyranose with (1→6)-β-D-galactopyranose branches locating at O-6 of mannose [11]. The difference on the glycosidic linkages partially determined the biological properties of natural polysaccharides. The FT-IR results in the present study also showed the typical functional groups of carbohydrates in Cs-4 EPSp and IPSp. However, the high-pressure gel permeation chromatography (HPGPC) profiles showed that both EPSp and IPSp contained two main fractions with different molecular weights, and further studies on specific fraction separation and structural elucidation were needed in the future.

A few studies reported the direct effects of several natural polysaccharides on platelets, such as fucoidan [22]. The results in the present study also showed that the two purified Cs-4 polysaccharides significantly inhibited the activation and aggregation of human platelets directly in vitro, and attenuated the thrombus formation in vivo. A previous study also reported that a polysaccharide CME-1 extracted from the mycelial of *Cordyceps sinensis* (strain number unannotated) inhibited human platelet activation [11]. However, many polysaccharides from plant- or fungus-based foods were resistant to gastrointestinal digestion and fermentable by gut bacteria in colon [23]. A lot of studies suggested that indigestible fibers exhibit partial biological functions through their gut microbial metabolites that passed through the colonic epithelium into the bloodstream [23]. SCFAs were the key bacterial metabolites of fermentable polysaccharides [23]. In our previous studies, the SCFA production was significantly increased after fecal fermentation of a high MW polysaccharides from a medical fungus Cs-HK1 [20]. Additionally, the SCFA concentration was negatively correlated to the MW of Cs-HK1 polysaccharides [21]. Therefore, the microbial metabolites of polysaccharides contributed a considerable proportion of the healthy benefits of the original polysaccharides in vivo.

The results showed that the inhibiting effects of the two Cs-4 polysaccharides on platelets were enhanced dramatically after fermentation with gut bacteria, which is consistent with a previous study that the anti-inflammatory activity of an exopolysaccharide from Cs-HK1 (a medicinal fungus) was also effectively enhanced by bifidobacterial fermentation [13]. Furthermore, the enhanced bioactivity was associated with the SCFA production and modification of polysaccharide structure [13]. The effects were probably attributed to the SCFAs produced by gut bacteria after fermentation with polysaccharides. Another study reported that trimethylamine-N-oxide (TMAO), which is a gut microbial metabolite from dietary phosphatidylcholine or L-carnitine, played important roles on the human blood platelet functions [24]. All the above information suggested that the gut microbial metabolites contributed to a considerable proportion of the health benefits (platelet-inhibiting effects in the present study) brought by fermentable polysaccharides. However, to the best of our knowledge, no studies reported the influence of gut bacterial fermentation on the platelet inhibitory effects of polysaccharides.

SCFAs were the main metabolites of gut microbiota with fermentation of polysaccharides [25]. The most abundant SCFAs generated by gut bacteria with fermentation of polysaccharides include acetic acid, propionic acid, and butyric acid [25]. The results in the supplementary data showed that acetic acid, propionic acid, and butyric acid significantly inhibited the platelet aggregation (Appendix A) and activation (Appendix A) induced by collagen, and attenuated the thrombus formation in mice (Appendix A). An in vitro study reported that the aged vinegar and acetic acid significantly inhibited platelet aggregation induced by several agonists [26]. Another study also reported that dietary butyric acid was negatively correlated with platelet aggregation in Japanese women [27]. These previous studies suggested that the gut microbial metabolites (SCFAs) of polysaccharides were closely related to platelet functions, which is consistent with the present results. However, as the complexity of in vivo gastrointestinal environment is much higher than in vitro fecal bacterial fermentation, further animal studies were needed to verify the inhibitory effects of Cs-4 polysaccharides and their gut microbial metabolites on platelet functions.

## 4. Materials and Methods

### 4.1. Chemicals

The monosaccharide standards (fucose, arabinose, glucose, mannose, galactose, xylose, and ribose) were from Sigma-Aldrich (St. Louis, MO, USA) and the SCFA standards (acetic acid, propionic acid, *n*-butyric acid, *i*-butyric acid, and valeric acid) were from Aradin (Shanghai, China). The various dextran standards with a molecular weight of 1–670 kDa were from International Laboratory USA (San Bruno, CA, USA). Appendix A presents all compounds used in this work with information from NCBI PubChem compound database.

### 4.2. Production of Cs-4 Polysaccharides

Cs-4 mycelial fermentation powder was purchased from the National Institutes for Food and Drug Control (Beijing, China). Figure 7 shows the production process of Cs-4 polysaccharides. Ten grams of Cs-4 powder was suspended in 100 mL of Milli-Q water with vigorous stirring for 1 h. The suspension was centrifuged at 4000× *g* (BECKMAN COULTER, Allegra X-30, rotor C0650) for 20 min to separate the fermentation supernatant and the mycelia. The supernatant was precipitated by 4 V of ethanol (EtOH, 95%, *v*/*v*) for overnight at 4 °C followed by centrifugation (BECKMAN COULTER, Allegra X-30, rotor C0650, 10,000× *g*, 20 min) and freeze dry to obtain the exopolysaccharides (EPS). The precipitate containing mycelia was re-suspended in 100 mL of Milli-Q water and treated by ultrasound (120 W, 40 kHz) at 80 °C for 1 h, followed by centrifugation (BECKMAN COULTER, Allegra X-30, rotor C0650, 4000× *g*, 20 min) to separate the supernatant containing water-soluble intracellular polysaccharides. Then, the supernatant was precipitated by 4 V of EtOH for overnight at 4 °C followed by centrifugation (BECKMAN COULTER, Allegra X-30, rotor C0650, 10,000× *g*, 20 min) and freeze dry to obtain the intracellular polysaccharides (IPS). EPS and IPS were further deproteinized by Sevage method [28] and freeze dried to obtain the purified EPS (EPSp) and IPS (IPSp).

### 4.3. Analysis of Chemical Composition and Molecular Weight

The total protein content of EPS, IPS, EPSp, and IPSp was determined by the Bradford method, using the bovine serum albumin (BSA, BioFroxx, Guangzhou, China) as the standard [29]. The total carbohydrate content of all samples was determined by the total carbohydrate assay kit (Cat#BC2710, Solarbio, Shanghai, China) according to the instructions.

Molecular weight (MW) of EPS, IPS, EPSp, and IPSp was measured by an HPGPC instrument equipped with a Waters 1525 HPLC system and two consecutive TSK-GEL columns (G5000 PWXL and G3000 PWXL, 7.8 × 300 mm) (Waters Co, Milford, MA, USA) as previously reported [30]. All samples were dissolved in Milli-Q water (5 mg/mL) and centrifuged at 6000 rpm (EPPENDORF, 5427 R, rotor FA-45-30-11) for 15 min. The supernatant was collected and filtered through a 0.45 μM membrane prior to injection. The calibration curve was obtained using dextran MW standards with the peak MW of 1, 5, 12, 25, 50, 80, 270, 410, and 670 kDa.

### 4.4. Methylation Analysis

Methylation was performed to investigate the glycosidic linkages of EPSp and IPSp accordingly to a previously reported method [31] with minor modifications. The sample (10 mg) was dissolved in 500 μL of dimethyl sulfoxide, and 1 mg of NaOH was added followed by incubation for 30 min. Then 50 μL of iodomethane solution was added and reacted for 1 h. One milliliter of Milli-Q water and 2 mL of dichloromethane were added followed by vortex mixing. Then the solution was centrifuged and the aqueous phase was discarded. This step was repeated for 3 times, and the organic phase was collected followed by drying. Then the sample was hydrolyzed by 100 μL of 2 M trifluoroacetic acid (TFA) at 121 °C for 90 h and dried at 30 °C with a rotary evaporator. Then 50 μL of 2 M ammonia and 50 μL of 1 M NaBD_4_ were added and reacted for 2.5 h. The reaction was stopped by 20 μL of acetic acid, and the solution was dried by N_2_ flow. The sample was washed by 250 μL of methanol and dried, and repeated for two times. Then 250 μL of acetic anhydride was added and reacted for 2.5 h. One milliliter of water was added and let to stand for 10 min. Five hundred microliters of dichloromethane was used for extraction and this step was repeated for three times. Finally, the dichloromethane phase was collected for gas chromatography-mass spectrometry (GC-MS) determination.

Methylation analysis was carried out by a GC-MS system consisted of an Agilent 7890A GC system and an Agilent 5977B quadruple MS system (Santa Clara, CA, USA) equipped with an Agilent G4567A autosampler (Santa Clara, CA, USA), a BPX70 column (25 m × 0.22 mm × 0.25 μm, Trajan Scientific and Medical, VIC, AUS) and an electron bombardment ion source. The initial column temperature was 140 °C, then raised to 230 °C at a rate of 3 °C/min and maintained for 3 min. The injection volume was 1.0 μL, and the split ratio was at 10:1. Helium was used as a carrier gas with a flow rate at 1.0 mL/min. The mass scanning range was 30–600 (*m*/*z*). The data were collected and analyzed by MassHunter workstation (Agilent, Santa Clara, CA, USA).

### 4.5. Analysis of Monosaccharide Composition by High-Performance Anion-Exchange Chromatography-Pulsed Amperometric Detector (HPAEC-PAD)

A previously reported method [32] with some modifications was used to analyze monosaccharide composition of Cs-4 polysaccharides. Analytical standards (glucose, mannose, galactose, rhamnose, arabinose, fructose, ribose, xylose, and fucose) for monosaccharide determination were purchased from Sigma (St. Louis, MO, USA). The stock solutions of each standard were prepared at a concentration of 10 mg/mL using Milli-Q water. Working mix standard solutions were prepared by diluting the stock solution with Milli-Q water to 0.4–40 μg/mL. All solutions were stored at 4 °C until use.

The Cs-4 polysaccharide (5 mg) was hydrolyzed with 2M TFA at 121 °C for 2 h, and dried by N_2_ flow. Then the sample was washed and dried by N_2_ flow repeatedly for 3 times. Finally, the residue was re-dissolved in Milli-Q water and filtered through a 0.22 μm membrane before injection.

An HPAEC Dionex ICS 5000 system equipped with a CarboPac™ PA20 anion-exchange column (150 × 3.0 mm, 10 μm; Dionex, Thermo Fisher, Waltham, MA, USA) and a PAD were used for monosaccharide composition analysis. Mobile phase A: 0.1 M NaOH. Mobile phase B: 0.1 M NaOH, 0.2 M NaAc. Monosaccharide elution was accomplished by a linear gradient from 5% to 20% (*v*/*v*) mobile phase B in 30 min at a flow rate of 0.5 mL/min. The injection volume was 5 μL, and the column temperature was maintained at 30 °C. The ICS5000 (Thermo Scientific) software and chromeleon 7.2 CDS (Thermo Scientific) were used to collect and process data, respectively.

### 4.6. FT-IR and SEM

The IR spectra of EPSp and IPSp were recorded by KBr disk method using an FT-IR spectrometer Vertex 70 (Bruker Co., Karlsruhe, Germany). The spectra from 4000 to 400 cm^−1^ were recorded according to a previous method [20]. All spectra were baseline corrected.

The morphological structures of EPSp and IPSp were observed using a field emission scanning electron microscope JSM-6330F (JEOL Co., Tokyo, Japan) as previously reported [20]. Based on the preliminary results, 2 mL of 5 mg/mL of the Cs-4 polysaccharide solutions were immediately frozen in liquid nitrogen, followed by lyophilization. The electron microscopy was set at 15.0 kV accelerating voltage.

### 4.7. Preparation of EPSp and IPSp Gut Bacterial Metabolites by In Vitro Fecal Fermentation

The fecal fermentation of EPSp and IPSp was performed according to our previous study [20]. Fresh feces were collected from a healthy donor, who have no digestive diseases and chronic metabolic disease, and had not received any antibiotic treatment or microecological modulators (probiotic, prebiotic, and synbiotics) during the past three months prior to the sample collection. Stools were collected in sterilized centrifuge tubes and stored in an anaerobic jar with an anaerobic gas generating sachet (Mitsubishi Gas Chemical Co., Inc., Tokyo, Japan), and used within half an hour. The fecal samples were diluted in ten-fold (*w*/*w*) of sterilized phosphate buffered saline (PBS) (pH 7.2–7.4). The homogenized suspension was centrifuged (BECKMAN COULTER, Avanti J-15R, JS-4.750, 500 rpm, 5 min) to obtain the bacterial suspension for the in vitro fermentation.

The basal medium consisted of peptone water (2 mg/mL), yeast extract (2 mg/mL), sodium bicarbonate (2 mg/mL), L-cysteine (0.5 mg/mL), bile salts (0.5 mg/mL), NaCl (0.1 mg/mL), hemin (0.05 mg/mL), dipotassium phosphate (0.04 mg/mL), monopotassium phosphate (0.04 mg/mL), calcium chloride hexahydrate (0.01 mg/mL), magnesium sulfate heptahydrate (0.01 mg/mL), Tween-80 (2 mL/L), phylloquinone (vitamin K1, 10 μL/L), resazurin (1 mg/L), and distilled water [33]. EPSp and IPSp were dissolved in the basal medium at the concentration of 5 mg/mL with vigorous stirring overnight. The pH was adjusted to 6.8 followed by autoclaving at 121 °C for 20 min. Each tube was inoculated with 10% (*w*/*w*) freshly prepared fecal slurry. All tubes were enclosed in anaerobic jars, and cultured in an incubator with shaking at 200 rpm and 37 °C for 24 h. The samples were centrifuged at 6000× *g* (EPPENDORF, 5427 R, rotor FA-45-30-11) for 15 min, and the supernatants (EPSpm and IPSpm) were used for SCFA determination and platelet experiments.

### 4.8. Healthy Subjects for In Vitro Experiment

Thirty healthy adults (25–40 years old) who had not taken any antiplatelet drugs, tea, coffee, nutritional supplements (including prebiotics and probiotics) within the previous 14 days were recruited for the in vitro platelet studies. Subjects with a history of serious medical conditions, such as hemostatic disorders, CVDs, hypertension, diabetes mellitus, or thyroid disorders, were excluded. This study was approved by the Ethics Committee of Sun Yat-sen University in China [(2019) No. 134], and written informed consent was obtained from all subjects in accordance with institutional guidelines and the Declaration of Helsinki before enrollment.

### 4.9. Human Gel-Filtered Platelets Preparation

Whole blood from healthy volunteers was drawn into a tube containing 3.8% sodium citrate (1/9, *v*/*v*) via venipuncture. PRP and gel-filtered platelets were prepared according to a previous report [34]. PRP was obtained by centrifugation at 1000× *g* (BECKMAN COULTER, Avanti J-15R, rotro JS-4.750) for 10 min at 25 °C, and then transferred to a new tube and placed in a 37 °C water bath before the experiment. Gel-filtered platelets were then isolated from the PRP by using a Sepharose 2B column in PIPES buffer (5 mM Piperazine-1,4-bisethanesulfonic acid, PIPES, 4 mM potassium chloride, 1.37 mM sodium chloride, 0.1% (*w*/*v*) glucose, pH 7.0).

### 4.10. Quantification of Lactate Dehydrogenase (LDH) Release

The possible cytotoxic effect of Cs-4 polysaccharides and their fecal bacterial metabolites on human platelets in vitro was assessed by measuring the LDH leakage from the cytosol using a kit (Nanjing JianCheng Bioengineering Institute, Nanjing, China) according to the manufacturer’s instructions. Human gel-filtered platelets (1 × 10^8^ cells per mL) were incubated with EPSp and IPSp (final concentration of 1 mg/mL), EPSpm and IPSpm (1 mg/mL), SCFAs (20, 40 and 60 μg/mL) or different controls (PBS, basal medium before and after fecal fermentation) for 40 min at 37 °C. The cell-free supernatant was obtained by centrifugation at 12,000× *g* (EPPENDORF, 5427 R, rotor FA-45-30-11) for 5 min to measure the LDH leakage at 490 nm. 1% Triton was used as a positive control in which cell death rate was normalized to 100%.

### 4.11. Assessment of Platelet Aggregation

As previously reported [34], a Chronolog aggregometer (Chrono-Log Corp., Havertown, PA, USA) was used to assess the influence of Cs-4 polysaccharides and their gut bacterial metabolites on platelet aggregation at 37 °C with a stirring speed of 1000 rpm. Human PRP or gel-filtered platelets (2.5 × 10^8^ cells per mL) were pre-incubated with EPSp, IPSp, EPSpm, and IPSpm (5 mg/mL or 1 mg/mL, 4:1, *v*/*v*) or different controls (PBS, basal medium before and after fecal fermentation, 1:4, *v*/*v*) for 40 min at 37 °C. Platelet aggregation was stimulated by collagen (2 μg/mL). The light transmission was monitored and documented for at least 6 min. PBS was used as the negative control (Ctl). The basal medium before fermentation was used as the medium control (Unctlm), and the basal medium without PS after fermentation was used as the fermentation control (Ctlm). All the data were collected and processed by Aggrolink8 software.

### 4.12. Detection of Platelet Activation by Flow Cytometry

To assess the platelet activation, the CD62P expression and PAC-1 binding to platelets were determined as previously reported [34] with slight modification. Gel-filtered platelets (1 × 10^6^ cells per mL) prepared from the blood of healthy subjects were incubated with EPSp, IPSp, EPSpm, and IPSpm (5 mg/mL or 1 mg/mL, 4:1, *v*/*v*) or different controls (PBS, basal medium before and after fecal fermentation, 1:4, *v*/*v*) for 10 min at 37 °C, followed by co-incubation for another 30 min with fluorescein isothiocyanate (FITC)-conjugated anti-human CD62P and FITC-conjugated PAC-1 antibodies, respectively. Then, collagen was added to activate platelets for 1 min in the presence of 1 mM Ca^2+^. Samples were immediately fixed with 1% paraformaldehyde, and analyzed using a CytoFLEX flow cytometer (Beckman Coulter Inc., Brea, CA, USA) within two hours. The data were collected and processed by CytExpert 2.0 software (Beckman Coulter Inc., Brea, CA, USA).

### 4.13. Tail Bleeding Time Test

Male C57BL/6J mice (aged 6–8 weeks old) purchased from Sun Yat-sen University were injected intravenously via the tail vein with 0.2 mL of EPSp (5 mg/mL), IPSp (5 mg/mL), EPSpm (5 mg/mL), IPSpm (5 mg/mL), PBS (Ctl), fecal fermentation basal mediums before (Unctlm) and after fecal fermentation (Ctlm). The doses of different interventions in the animal study were calculated according to the doses used in the in vitro experiments, and finally achieved approximately 1 mg/mL in the murine blood. Mice were placed on a heating pad (37 °C), and anesthetized by sodium pentobarbital after 40 min. Five millimeters of the tail tip was cut off, followed by immediately placing the tail into a warm (37 °C) saline solution. The bleeding time was recorded as the period from the moment that bleeding began to the moment that bleeding ceased [35]. All experiments were approved by the Animal Care and Use Committee of Sun Yat-sen University.

### 4.14. Intravital Microscopy of Thrombosis Induced by FeCl_3_ In Vivo

Intravital microscopy was conducted according to our previous reports with minor modification [34,36]. Male C57BL/6J mice were obtained from Sun Yat-sen University. Animal experiments were approved by the Ethics Committee of Sun Yat-sen University [No. SYXK [Yue] 2017-0080].

Blood was drawn from the anesthetized donor-matched mice (6–8 weeks) following cardiac puncture into a syringe containing 10% volume of 3.8% sodium citrate. PRP was obtained by centrifugation at 1250× *g* (BECKMAN COULTER, Avanti J-15R, rotor JS-4.750) for 7 min. Gel-filtered platelets were prepared as described in Section 4.9 and concentrated to 200 μL, followed by incubation with 4 μg/mL Calcein AM (Sigma, MO, USA), and different interventions (5 mg/mL of EPSp, IPSp, EPSpm and IPSpm, 1:4, *v*/*v*) or controls (PBS, fecal fermentation basal mediums before and after fecal fermentation, 1:4, *v*/*v*) for 40 min at 37 °C for 40 min. Then the labeled platelets (4–5 × 10^9^ cells per kg of body weight, bw) were injected into the recipient mice (3–4 weeks old) via the tail vein. The recipient mice were then immediately anesthetized with pentobarbital sodium, and the mesentery was externalized. Mesenteric arterioles of 80 to 100 μm diameter were injured by a Whatman No. 1 paper (2 mm × 1 mm) saturated with 10% (*w*/*v*) FeCl_3_ solution for 2 min, followed by continuously recording for another 40 min by a Nikon SMZ25 Intravital Microscopy System (Tokyo, Japan). The thrombus formation at different time points was compared among different intervention groups and control groups. The pictures were processed and analyzed by NIS-Elements software.

### 4.15. Statistical Analysis

ANOVA followed by the least significant difference (LSD) or Tamhane’s T2 was applied to compare the SCFA concentration after fecal fermentation, the aggregation rate of platelets, the CD62p expression, and binding to PAC-1 antibody of platelets among different groups. Student’s *t*-test was used to compare the thrombus formation between the experimental group and the control group. Data are presented as the mean ± standard deviation (SD) of at least three independent experiments. *p* value < 0.05 indicated a significant difference. SPSS 16.0 and GraphPad Prism version 5.01 software (GraphPad Inc., San Diego, CA, USA) were used for statistical analysis.

## 5. Conclusions

The exopolysaccharides (EPSp) and intracellular polysaccharides (IPSp) isolated and purified from the Cs-4 mycelial fermentation powder are majorly composed of galactose, mannose, glucose, and arabinose. The two Cs-4 polysaccharides showed significant effect on inhibiting the activation and aggregation of human platelets induced by collagen and decreasing the CD62P expression and αIIbβ3 activation on platelets. Moreover, EPSp, IPSp, and their metabolites also inhibited thrombus formation in vivo without increasing bleeding risk. Notably, the anti-platelet effects of EPSp and IPSp were dramatically increased after fecal bacterial fermentation, which indicated the important role of gut microbiota in the health-promoting function of the two polysaccharides. To the best of our knowledge, this is the first study investigating the Cs-4 polysaccharides and their gut bacterial metabolites on platelet functions. The results suggested the potential effects of Cs-4 polysaccharides to prevent thrombotic diseases through inhibiting platelet functions. Further studies were needed to investigate the most effective fractions of Cs-4 polysaccharides and verify the animal feeding experiments.

## Figures and Tables

**Figure 1 ijms-23-12608-f001:**
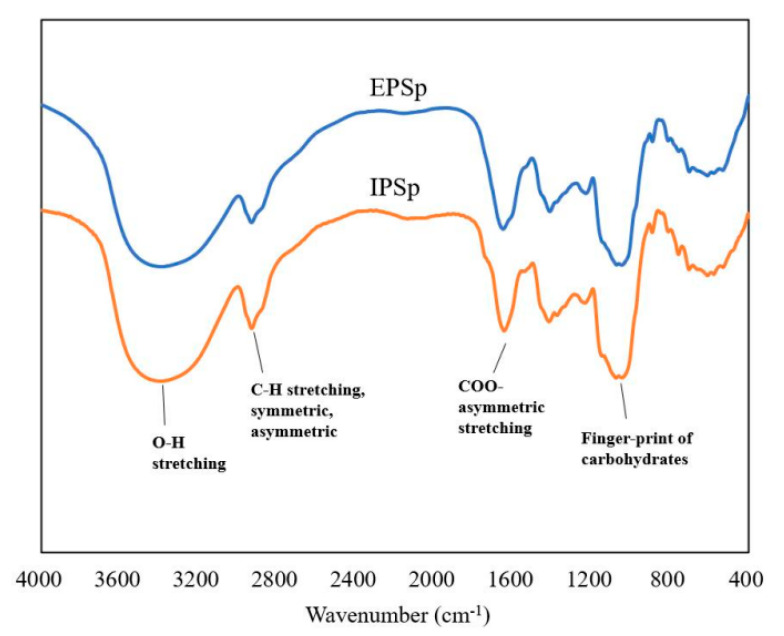
The FT-IR spectra of EPSp and IPSp. EPSp: purified Cs-4 exopolysaccharides, IPSp: purified Cs-4 intracellular polysaccharides.

**Figure 2 ijms-23-12608-f002:**
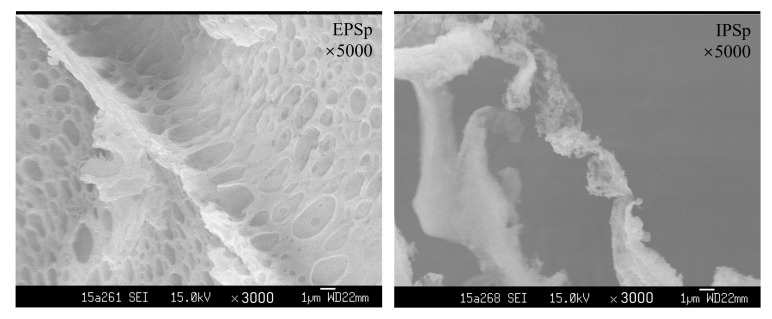
Scanning electron microscopy observation of EPSp and IPSp. EPSp: purified Cs-4 exopolysaccharides, IPSp: purified Cs-4 intracellular polysaccharides.

**Figure 3 ijms-23-12608-f003:**
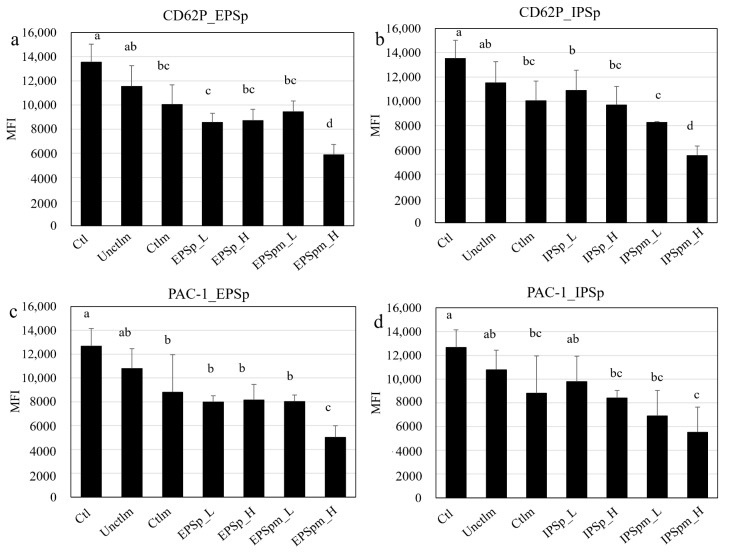
The effects of Cs-4 EPSp, IPSp, and their gut bacterial metabolites on human platelet activation. (**a**) CD62P expression with EPSp; (**b**) CD62P expression with IPSp; (**c**) PAC-1 binding to platelets with EPSp; (**d**) PAC-1 binding to platelets with IPSp; Ctl: phosphate buffered saline; Unctlm: fecal fermentation basal medium before fecal fermentation; Ctlm: fecal fermentation basal medium after fecal fermentation; EPSp: purified Cs-4 exopolysaccharides, IPSp: purified Cs-4 intracellular polysaccharides. EPSpm: EPSp after fecal fermentation; IPSpm: IPSp after fecal fermentation. H: 5 mg/mL for EPSp, IPSp, EPSpm and IPSpm; L: 1 mg/mL for EPSp, IPSp, EPSpm and IPSpm. Collagen: 2 μg/mL. Data are presented as mean ± standard deviation, n = 3. Different letters indicate significant difference among different groups for the same SCFA, *p* < 0.05, ANOVA.

**Figure 4 ijms-23-12608-f004:**
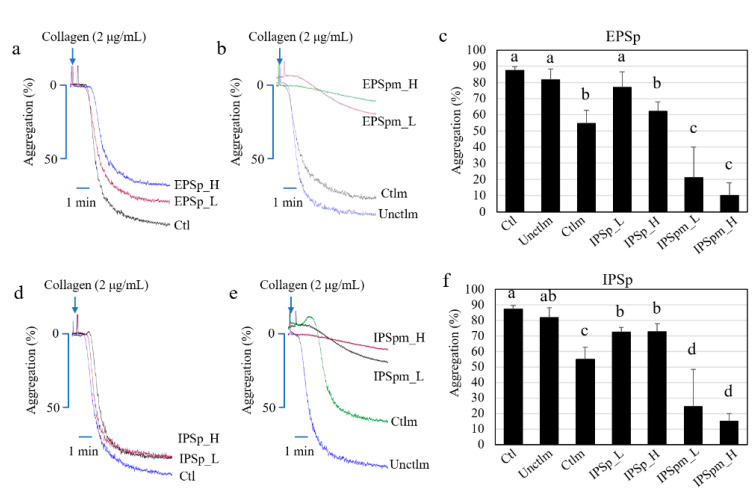
The effects of Cs-4 EPSp, IPSp, and their gut bacterial metabolites on human platelet aggregation induced by collagen. (**a**) EPSp; (**b**) EPSpm; (**c**) bar chart of EPSp and EPSpm; (**d**) IPSp; (**e**) IPSpm; (**f**) bar chart of IPSp and IPSpm. Ctl: phosphate buffered saline; Unctlm: fecal fermentation basal medium before fecal fermentation; Ctlm: fecal fermentation basal medium after fecal fermentation; EPSp: purified Cs-4 exopolysaccharides, IPSp: purified Cs-4 intracellular polysaccharides. EPSpm: EPSp after fecal fermentation; IPSpm: IPSp after fecal fermentation. H: 5 mg/mL for EPSp, IPSp, EPSpm and IPSpm; L: 1 mg/mL for EPSp, IPSp, EPSpm and IPSpm. Collagen: 2 μg/mL. Data are presented as mean ± standard deviation, n = 3. Different letters indicate significant difference among different groups for the same SCFA, *p* < 0.05, ANOVA.

**Figure 5 ijms-23-12608-f005:**
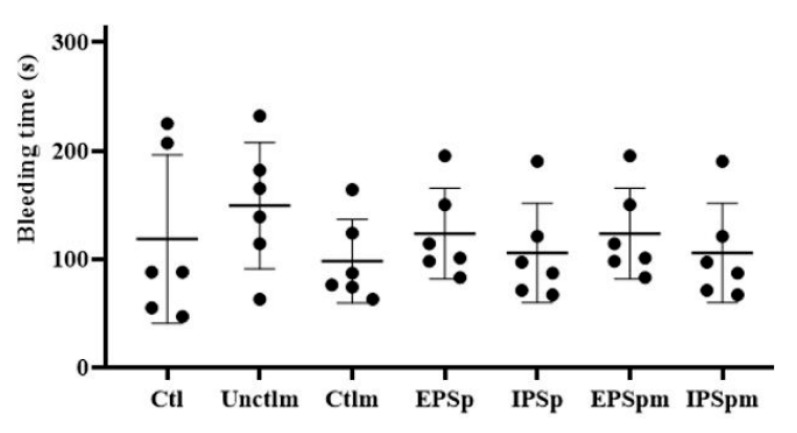
The effects of Cs-4 EPSp, IPSp, and their gut bacterial metabolites on bleeding time in C57BL/6J mice. Ctl: phosphate buffered saline; Unctlm: fecal fermentation basal medium before fecal fermentation; Ctlm: fecal fermentation basal medium after fecal fermentation; EPSp: purified Cs-4 exopolysaccharides, IPSp: purified Cs-4 intracellular polysaccharides. EPSpm: EPSp after fecal fermentation; IPSpm: IPSp after fecal fermentation. Each black dot represents a mouse. Data are presented as mean ± standard deviation, n = 6. *p* < 0.05, ANOVA.

**Figure 6 ijms-23-12608-f006:**
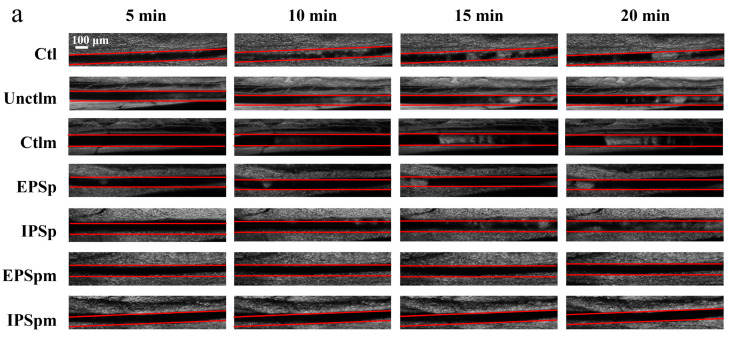
The effects of Cs-4 EPSp, IPSp, and their gut bacterial metabolites on thrombus formation induced by FeCl_3_ in C57BL/6J mice. (**a**): Presentative pictures of mesenteric arterioles at different time points, and the red lines indicate the mesenteric arterioles; (**b**): time to 1st thrombus (>20 μm); (**c**) number of thrombus (>20 μm); (**d**) size of thrombus over 2 min starting from the time of the first thrombus (>20 μm). Ctl: phosphate buffered saline; Unctlm: fecal fermentation basal medium before fecal fermentation; Ctlm: fecal fermentation basal medium after fecal fermentation; EPSp: purified Cs-4 exopolysaccharides, IPSp: purified Cs-4 intracellular polysaccharides. EPSpm: EPSp after fecal fermentation; IPSpm: IPSp after fecal fermentation. Each black dot represents a mouse. Data are presented as mean ± standard deviation, n = 6. * *p* < 0.05, student’s *t*-test, compared to the control group.

**Figure 7 ijms-23-12608-f007:**
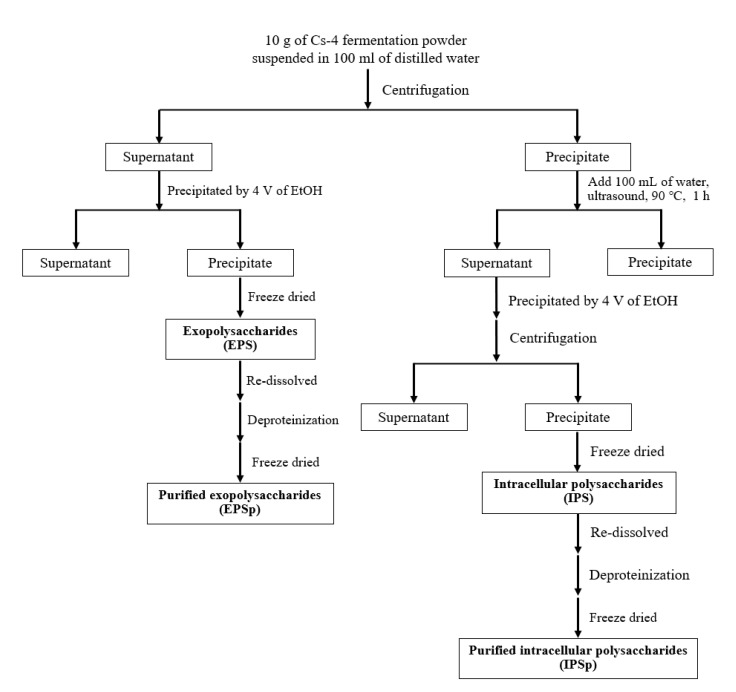
Production process of Cs-4 polysaccharides.

**Table 1 ijms-23-12608-t001:** Yields and molecular properties of EPS, IPS, EPSp, and IPSp.

	Yields (%)	MW (kDa, Percentage of Peak Area)	Carbohydrate (%)	Protein (%)
EPS	6.95	41.9, 75.07%	88.08 ± 2.07	3.93 ± 0.46
		0.8, 24.35%		
IPS	3.65	26.8, 78.35%	84.00 ± 1.94	4.57 ± 0.38
		1.1, 20.73%		
EPSp	2.87	32.7, 86.05%	90.09 ± 4.74	1.11 ± 0.01
		1.05, 13.95%		
IPSp	0.75	28.2, 20.41%	88.90 ± 0.41	0.78 ± 0.38
		2.85, 54.38%		

EPS: exopolysaccharides, EPSp: purified exopolysaccharides, IPS: intracellular polysaccharides, IPSp: purified intracellular polysaccharides. For carbohydrate and protein determination, data were shown in average ± standard deviation, n = 3.

**Table 2 ijms-23-12608-t002:** Monosaccharide contents and molar ratios of EPSp and IPSp.

	Monosaccharide Contents (μg /mg)		Molar Ratio of Monosaccharide	
	EPSp	IPSp	EPSp	IPSp
Fuc	5.01	7.48	0.01	0.02
Ara	38.13	47.42	0.10	0.12
Gal	177.68	159.15	0.37	0.33
Glc	102.45	131.56	0.22	0.27
Xyl	12.45	16.76	0.03	0.04
Man	121.77	110.40	0.26	0.23
Rib	5.37	2.08	0.01	0.01

**Table 3 ijms-23-12608-t003:** Short chain fatty acid (SCFA) production (mM) after fecal fermentation with Cs-4 EPSp and IPSp (5 mg/mL). Unctlm: fecal fermentation basal medium before fecal fermentation. Ctlm: fecal fermentation basal medium after fecal fermentation; EPSpm: EPSp after fecal fermentation; IPSpm: IPSp after fecal fermentation. n = 3, ANOVA, different letters indicate significant difference among different groups for the same SCFA, *p* < 0.05.

	Acetic Acid	Propionic Acid	Butyric Acid	Valeric Acid	Total SCFAs
Unctlm	5.13 ± 0.39 c	2.10 ± 0.99 b	0.57 ± 0.08 c	0.26 ± 0.17 c	8.06 ± 1.27 c
Ctlm	7.23 ± 0.22 b	2.73 ± 0.80 b	1.58 ± 0.09 b	1.19 ± 0.06 b	12.73 ± 0.69 b
EPSpm	13.01 ± 0.05 a	10.51 ± 1.29 a	4.25 ± 0.23 a	1.54 ± 0.25 a	29.31 ± 1.78 a
IPSpm	13.96 ± 1.00 a	10.69 ± 0.16 a	3.89 ± 0.27 a	1.20 ± 0.10 a	29.74 ± 0.60 a

Data were shown in average ± standard deviation.

## Data Availability

Not applicable.

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
