# Peer review of "Extraction, Characterization, and Platelet Inhibitory Effects of Two Polysaccharides from the Cs-4 Fungus"

_ijms, 2022, doi:10.3390/ijms232012608_

Round 1

Reviewer 1 Report

The manuscript presents large and comprehensive work on characterization of Cs-4 polysaccharides composition and biological activity which is acceptable for publication 

minor comments

1. It would be better to specify which methods were used to determine the protein and polysaccharide contens in isolates, as well as molecular weights of polysaccharides under study

2. The figures quality could be better regarding image resolution

3. Why didn't authors use pure strain for polysaccharides fermentation?

4. Is it possible the Cs-4 polysacchrides are sulfated like fucoidan which is also has platelet aggregation inhibitory activity?

in figure 4 caption the word "platelet" is missing "Figure 4. The effects of Cs-4 EPSp, IPSp, and their gut bacterial metabolites on human activation. (a)..."

Reviewer 2 Report

see file

Round 2

Reviewer 2 Report

see file
